Nicotine smoking is associated with impaired cognitive performance in Pakistani young people

Riaz Tuba 1
Murtaza Ghulam drghulam.murtaza@uog.edu.pk 1
Arif Areej 1
Mahmood Shahid 1
Sultana Razia 2
Al-Hussain Fawaz 3
Bashir Shahid 4
1 Department of Zoology, University of Gujrat , Gujrat , Pakistan
2 Institute of Agricultural Sciences, University of the Punjab , Lahore , Pakistan
3 College of Medicine, Department of Neurology King Saud University , Riyadh , Saudi Arabia
4 Neuroscience Center, King Fahad Specialist Hospital , Dammam , Saudi Arabia
Chen Hao
Electronic publication date: 2021 Jun 2
Publication date: 2021
Volume: 9
Electronic Location ID: e11470
Received 2020 Oct 7; Accepted 2021 Apr 26
Copyright: ©2021 Riaz et al.
Copyright year: 2021
Copyright holder: Riaz et al.
License: This is an open access article distributed under the terms of the Creative Commons Attribution License, which permits unrestricted use, distribution, reproduction and adaptation in any medium and for any purpose provided that it is properly attributed. For attribution, the original author(s), title, publication source (PeerJ) and either DOI or URL of the article must be cited.
License URL: https://creativecommons.org/licenses/by/4.0/

Keywords: Nicotine smoking, Attention switching task (AST), Pattern recognition memory (PRM), Choice reaction time (CRT), Cambridge neuropsychological automated battery (CANTAB)

Funding: The authors received no funding for this work.

==============================
Nicotine smoking is the most common mode of tobacco smoking among young people. It affects the areas of the brain associated with memory, attention, and learning. This study has investigated the effect of nicotine smoking on cognitive performance. One hundred male volunteers (50 nicotine smokers and 50 nonsmokers) aged 18–30 years with similar socioeconomic backgrounds were recruited for this study. Clinical history of participants was obtained using a questionnaire. Their brain health and handedness were determined using the Mini Mental State Examination (MMSE) and the Edinburgh Handedness Inventory (EHI), respectively. The dependent variables examined in the study were attention-switching tasks (AST), pattern recognition memory (PRM), and choice reaction time (CRT). These parameters were assessed using the Cambridge Neuropsychological Automated Battery (CANTAB). The average ages of participating smokers and nonsmokers were 24.02 ± 3.41 years (mean ± standard deviation) and 22.68 ± 1.87 years, respectively. MMSE and EHI scores of smokers were 28.42 ± 1.09 and 99.75 ± 1.77, respectively; for nonsmokers, these scores were 28.54 ± 1.34 and 98 ± 1.91, respectively. The mean score for AST correct latency/delay was significantly higher (p = 0.050) in smokers (620.26 ± 142.03) than in nonsmokers (570.11 ± 108.39). The percentage of correct AST trials was significantly higher (p = 0.000) in nonsmokers (96.95 ± 2.18) than in smokers (83.75 ± 11.22). The PRM percent correct were significantly higher (p = 0.000) of nonsmokers (93.42 ± 8.34) than of smokers (79.75 ± 13.44). The mean correct latency for CRT was significantly higher (p = 0.009) in smokers (509.87 ± 129.83) than in nonsmokers (455.20 ± 65.73). From this data, it can be concluded that nicotine smoking is linked with impaired cognitive functions in smokers.

Introduction

A cigarette is a well-engineered device to inhale smoke deep into the lungs (Benjamin, 2011; Centers for Disease Control and Prevention (US), National Center for Chronic Disease Prevention and Health Promotion (US), Office on Smoking and Health (US), 2010). In the USA, in 2016, prevalence of cigarette smoking was 17.5% and 13.5% among males and females, respectively (Jamal et al., 2018). In Pakistan, age-standardized prevalence of tobacco consumption is 13.4% (Basit et al., 2020). According to the World Health Organization (WHO), tobacco use is the leading cause of poverty, disease, and death worldwide (WHO, 2020). It causes more than eight million deaths per year, 1.2 million deaths occur due to second-hand smoke (WHO, 2020). These possible consequences of cigarette smoking demonstrate its dangers, yet it is practiced by a considerable number of individuals (Benowitz, 2010). Nicotine dependence is considered a “substance use disorder” and is listed in the fourth edition of the Diagnostic and Statistical Manual of Mental Disorders (DSM-IV) (1994) and its text revised edition the DSM IV TR (2000) (American Psychiatric Association, 2000; Bell, 1994). Cigarette smokers perceive themselves to enjoy more at parties while smoking and experience more relaxation (Klein, Sterk & Elifson, 2014). Another reason smokers continue to smoke is that they may experience impaired working memory, attention, and concentration when they try to quit (Harrison, Coppola & McKee, 2009; McClernon et al., 2008).

The impact of nicotine on cognitive performance is an interesting and active area of research. Nicotine administration causes release of the neurotransmitter acetylcholine (Reid, Lloyd & Rao, 1999), which is associated with increase in attention (Herrero et al., 2008). Nicotine has significant positive effects on attention, memory, and motor abilities (Heishman, Kleykamp & Singleton, 2010). Alertness and attention are found better in mild smokers as compared to nonsmokers (Vajravelu et al., 2015). On the other hand, a prospective questionnaire study conducted in subjects recruited from Minnesota state of the U.S has described negative association between tobacco use and cognitive function (Ge et al., 2019). Recent studies have also demonstrated an association between smoking and cognitive impairment in Saudi (Bashir et al., 2017; Bashir et al., 2020) and in Chinese (Xia et al., 2019) adult populations. For clear demonstration of association between nicotine smoking and cognitive function, the Cambridge Neuropsychological Test Automated Battery (CANTAB) was used in this study. The CANTAB is a standard software used to assess cognitive functions such as attention, memory, and decision-making ability (Chamberlain et al., 2012; Karlsen et al., 2020). As, it is a computer-based test, it requires less time for completion. Results are not affected by tiredness of participants (Chamberlain et al., 2012; Karlsen et al., 2020). Moreover, it provides more accurate results as compared to paper-based cognitive assessment tests, particularly on tasks that require time counting and may involve response delay, such as attention switching task (AST).

Response to drugs varies in different populations (Edwards & Aronson, 2000). Previously, effect of nicotine smoking on cognitive function has been investigated in Chinese population (Xia et al., 2019), the US population (Chamberlain et al., 2012), and Saudi population (Bashir et al., 2017; Bashir et al., 2020) using CANTAB. Impact of nicotine smoking on cognitive performance remains controversial (Chamberlain et al., 2011). It would be of great interest to investigate the effect of nicotine smoking on cognitive function in other populations. Our ultimate aim was to investigate the association between nicotine smoking and cognitive performance in Pakistani male young population in the hope to clarify the role of nicotine on cognitive performance.

Materials and Methods

This cross-sectional study was conducted in the Department of Zoology, University of Gujrat, Punjab, Pakistan from February to May 2019. Socio-demographic and -economic status-matched design was employed to assess the association between cigarette smokers and nonsmokers cognitive function.

Selection of subjects

Selection of participants was based on their volunteer participation. Subjects were asked about their social, demographic, economic, and health characters through face-to-face interviews. Socio-demographic and -economic variables included age, gender (male), marital status (married and single), education (subjects were either pursuing their education after completing 14 years of education or have completed 16 years of education with no further study) and employment (unemployed and employed; employed subjects had monthly income less than 560 U.S Dollars. Subjects who had smoked at least five cigarettes per day for the last year were assigned to the smoking group. Nonsmokers (control group) had never smoked a cigarette in the last 30 days. Although, they might have used it (tobacco cigarette, e-cigarette, and waterpipe) in social gatherings in the past but never smoked on regular basis. Hundred male participants were recruited for this study. Each group of smokers and nonsmokers consisted of 50 individuals aged 24.02 ± 3.41 (mean ± SD) years and 22.68 ± 1.87 years, respectively. Participants’ ages ranged from 18 to 30 years. The sample size of the present study was based on the previous studies which demonstrated significant association between smoking and cognitive function impairment; smokers n = 22, nonsmokers n = 30 (Bashir et al., 2017), nicotine smokers, shisha smokers, and nonsmokers n = 25 each (Bashir et al., 2020). Thus, we planned to recruit 100 participants (smokers and nonsmokers n = 50 each), which would be needed to investigate the association between smoking and cognitive function. All the participants in this study were male as it was difficult to recruit female smokers for research due to cultural reasons. Moreover, smoking prevalence is higher in men than women in Pakistan (Basit et al., 2020).

Exclusion Criteria: Individuals addicted to alcohol or other addictive substances and suffering from diabetes mellitus, obesity, anemia, obstructive lung diseases, malignance, difficulty in vision, attention, psychiatric problems, seizures, musculo-skeletal disorders and disturbed sleep history were excluded from the study (Bashir et al., 2017; Bashir et al., 2020).

Ethical Approval: Verbal consent was obtained from participants for their inclusion in the study. All procedures were conducted according to the Declaration of Helsinki and ethical standards of local Institutional Review Board (IRB), University of Gujrat. Moreover, ethical approval was obtained from IRB prior to starting the study (Ref: UOG/ORIC/2019/326).

Cognitive function

Mini Mental State Examination (MMSE) (Folstein, Folstein & McHugh, 1975) and Edinburg Handedness Inventory (EHI) (Veale, 2014) were used to determine the mental health and dominant hand of participants, respectively.

We employed CANTAB research suit software, version 6.0.37 (Karlsen et al., 2020) to perform different tests, which included AST, pattern recognition memory (PRM), and choice reaction time (CRT) to assess cognitive function. The test takes 25 to 30 min to complete. During the test, participants sat comfortably in front of computer and pressed the response button. The testing process has been described in detail in a previous study (Bashir et al., 2017).

Attention switching task (AST)

The detail description of the AST has been described in a previous study (Bashir et al., 2020). This test assesses the participant’s ability to switch attention between the direction of an arrow and its location on the screen and to ignore task-irrelevant information in the face of interfering or distracting events. It measures the cognitive control processes involved in the prefrontal cortex part of the brain and determines the executive function. In this test, each trial displays a cue at the top of the screen that indicates to the participant whether they have to press the right or left button according to the “side on which the arrow appears” or the “direction in which the arrow is pointing”. Some trials display congruent stimuli (arrow is present on the right side of the screen pointing to the right) whereas other trials display incongruent stimuli, which require a higher cognitive demand (arrow is present on the right side of the screen pointing to the left). AST test outcome measures include response latencies and error scores.

Pattern recognition memory (PRM)

This is a task of visual pattern recognition memory. In this task, a paradigm design for the visual pattern is displayed on the screen to memorize it (Bashir et al., 2020). The visual patterns are not easy to be labelled verbally. Then, the paradigm is displayed again for the second time with pairs. The person has to select the correct pattern that he has already seen. PRM task allows to measure the number and percentage of correct patterns selected. In this test, PRM raw score obtained represents PRM percent correct which indicates the number of correct responses expressed as a percentage.

Choice reaction time (CRT)

CRT measures the reaction time for two stimuli displayed on the computer screen. These stimuli are with two choices either the arrow is present on the left side or right side. Reaction time (ms) is measured when a person presses the button for the left or right side. Using this test, correct and incorrect responses, latency (response speed), and errors of commission and omission (late and early responses) can be assessed.

Statistical analysis

SPSS software was used to analyze the data. The quantitative data of smokers and nonsmoker groups were compared. As two groups were unrelated, t-test for independence was used for comparison. Significance among groups was calculated by applying t-test for independence. P-values ≤0.05 were considered as significant. Mean difference and standard deviation were also calculated and compared in the present study.

Results

The study participants consisted of 100 male volunteers. Demographic data of all participants were collected using a questionnaire. Numbers of participants who had completed 14–16 years education, were married or single, and employed or unemployed were comparable between smokers and nonsmokers. The MMSE score of 24 or more (out of 30) indicates mental stability. Handedness was determined using the EHI to ensure that subjects used their dominant hand to complete all the tests. The MMSE scores of smokers and nonsmokers were 28.42 ± 1.09 and 28.54 ± 1.34, respectively. The EHI scores of smokers and nonsmokers were 99.75 ± 1.77 and 98 ± 1.91, respectively (Table 1).

Table 1 Demographic data of smokers and nonsmokers.

Parameters	Smokers	Nonsmokers	
Gender	Male	50	50	
Female	0	0	
Age (years)	Mean ± SD	24.02 ± 3.41	22.68 ± 1.87	
Education	Years of education	14-16	14-16	
Marital status	Married, No. (%)	9 (18)	14 (28)	
Single, No. (%)	41 (82)	36 (72)	
Employment status	Employed, No. (%)	25 (32)	21 (42)	
Unemployed, No. (%)	34 (68)	29 (58)	
MMSE	Mean ± SD	28.42 ± 1.09	28.54 ± 1.34	
EHI	Mean ± SD	99.75 ± 1.77	98 ± 1.91	
Notes.

MMSE mini mental state examination

EHI Edinburg handedness inventory

SD standard deviation

The AST congruent cost (mean, correct) and mean AST correct latency (block 7) (switching block) were higher for smokers than for nonsmokers. However, this difference was not significant. The mean AST correct latency (blocks 3, 5) (non-switching blocks) was also higher for smokers (574.95 ± 111.05) than for nonsmokers (501.56 ± 68.10); this difference was significant (p =0.000). The AST switching cost (mean, correct) did not differ significantly between smokers and nonsmokers. The mean AST correct latency was significantly higher (p =0.050) for smokers (620.26 ± 142.03) than for nonsmokers (570.11 ±  108.39). The mean AST correct latency (congruent) and AST correct latency (incongruent) were significantly higher for smokers (p =0.046, p =0.036, respectively) than for nonsmokers. The percentage of correct AST trials was significantly higher (p =0.000) for nonsmokers (96.95  ± 2.18) than for smokers (83.75 ± 11.22). The PRM percent correct of smokers were 79.75 ± 13.44 and of nonsmokers were 93.42 ± 8.34. This difference was also significant (p =0.000).

Mean CRT correct latency of smokers (509.87 ± 129.83) was significantly higher (p =0.009) than that of nonsmokers (455.20 ± 65.73). The percentage of correct CRT trials was lower for smokers (95.30 ± 4.09) than for nonsmokers (98.48 ± 1.64); this difference was also significant (p =0.000) (Table 2, Fig. 1).

Table 2 Comparison of cognitive functions (AST, PRM, and CRT) between smokers and nonsmokers.

Parameters	Smokers (Mean ± SD)	Nonsmokers (Mean ± SD)	p-value	
AST-Congruency cost	76.00 ± 57.49	65.37 ± 40.82	0.289	
AST-Latency (blocks 3,5) [non-switching blocks]	574.95 ± 111.05	501.56 ± 68.10	0.000	
AST-Latency (block 7) [switching block]	669.96 ± 208.13	640.94 ± 167.21	0.444	
AST-Switching cost	95.01 ± 166.39	139.37 ± 130.21	0.141	
AST-Latency	620.26 ± 142.03	570.11 ± 108.39	0.050	
AST-Latency (congruent)	586.50 ± 132.46	538.36 ± 104.00	0.046	
AST-Latency (incongruent)	662.50 ± 157.44	603.72 ± 116.56	0.036	
AST-Percent correct trials	83.75 ± 11.22	96.95 ± 2.18	0.000	
PRM- Percent correct	79.75 ± 13.44	93.42 ± 8.34	0.000	
CRT-Latency	509.87 ± 129.83	455.20 ± 65.73	0.009	
CRT-Percent correct trials	95.30 ± 4.09	98.48 ± 1.64	0.000	
Notes.

PRM-Percent correct are the number of correct responses, expressed as a percentage.

AST attention switching task

PRM pattern recognition memory

CRT choice reaction time

SD standard deviation

Significance among groups was calculated by applying t-test for independence. P-values ≤0.05 were considered as statistically significant.

Figure 1 Comparison of cognitive functions between smokers and nonsmokers.

Y-axis shows time taken by the participants to respond. X-axis represents different parameters investigated. AST = attention switching task, CRT = choice reaction time. Latency values are “mean correct”. The results from smokers and nonsmokers were compared by independent t-test. The error bar and * represent standard error of the mean (SEM) and significant difference, respectively.

Discussion

The present study examines the effect of nicotine smoking on cognitive function in young people of Gujrat, Punjab, Pakistan. Three functional outcomes—attention, memory, and reaction time—were assessed for participants in the same age group with the same levels of education and socioeconomic status. Our data indicate cognitive impairment in nicotine smokers. A previous study conducted in subjects from British Civil service reported that middle-aged smokers experienced faster age-related decline in cognitive function than nonsmokers (Sabia et al., 2012). In young adult smokers of Minneapolis-St. Paul area, Minnesota, USA, cognitive impairment was demonstrated in different domains such as spatial working memory, sustained attention, executive planning, and spatial working memory (Chamberlain et al., 2012). In recent studies (Bashir et al., 2017; Bashir et al., 2020), impairment in attention and PRM was demonstrated in young adult smokers of Saudi Arabia. Thus, our data are in concordance with previous findings (Bashir et al., 2017; Bashir et al., 2020).

Previous studies have found that cognitive changes occur with normal aging (Harada, Natelson Love & Triebel, 2013; Murman, 2015). Since age affects cognitive performance, we selected participants of the same age group for the present study. Research has found that poor socioeconomic status (Raizada & Kishiyama, 2010; Yang et al., 2016) and low levels of education can negatively affect cognitive performance (Lipina & Posner, 2012) An association between higher levels of education and higher cognitive function has been demonstrated in Lee et al. (2006), Zahodne, Stern & Manly (2015). Thus, to avoid any confounding effects of socioeconomic status, we recruited subjects with similar socioeconomic statuses.

We observed significant impairment to the cognitive function of smokers. Values of AST mean correct latency and AST mean correct latency congruent and incongruent were significantly higher for smokers than for nonsmokers. These results align with previous studies (Bashir et al., 2017; Bashir et al., 2020) of Saudi young people, where the software (CANTAB) and the parameters; AST and PRM were used for investigation. We also found that PRM percent correct of nonsmokers were significantly higher than of smokers. This was similar to a previous study Bashir et al. (2017), although the difference in that case was not significant. In another recent study (Bashir et al., 2020), PRM percent correct were higher for smokers as compared to nonsmokers, however results were not significantly different. These results were attributed to preservation of memory functions in smokers (Bashir et al., 2017). However, these differences in results can also be attributed to differences in sample size and the socioeconomic conditions of the sample groups, as age-related cognitive change affects people living under different socioeconomic conditions differently (Weng et al., 2018). A recent study (Xia et al., 2019) examined the effect of nicotine smoking on Chinese population (age 45–46 years) using repeatable battery for the assessment of neuropsychological status (RBANS) and demonstrated lower immediate and delayed memory scores in smokers compared to nonsmokers. We have also found significantly higher CRT latency values for smokers compared to nonsmokers, which suggests impairment in ability to maintain vigilance and attention for the target stimulus in smokers. To the best of our knowledge, this is the first study that demonstrates the cognitive effect of nicotine smoking using CRT parameters.

Our finding that there is an association between smoking and cognitive problems aligns with other studies (Durazzo, Meyerhoff & Nixon, 2012; Fried, Watkinson & Gray, 2006) in which smokers performed worse than nonsmokers on memory, processing speed, and visuospatial learning tasks assessed using a neurocognitive battery. In this study, we compared the cognitive function of smokers and nonsmokers using CANTAB, which is an automated, neuropsychological battery and may offer unbiased results for different cognitive functions such as attention, learning, memory, and executive functions (Collie et al., 2007; Karlsen et al., 2020; Robbins et al., 1994).

The limitations of our study are the small sample size and its cross-sectional design. Our data are from low- to middle-income subjects. High-income subjects, white-collar civil servants, and females were not included in this study. Moreover, this study is from a small area (Gujrat) and does not cover all the areas of Pakistan. Thus, we cannot say that these data are from general population. In our study, smoking behavior and health measures were self-reported. Number of cigarettes smoked in routine reported by smokers may vary. No laboratory test was performed to determine the addiction pattern and health measures in smokers. Larger prospective studies with more detailed assessments are required to identify the true links between smoking and cognitive impairment. In our study, we were unable to recruit female smokers. Moreover, higher smoking prevalence is demonstrated in men as compared to women in Pakistan (Basit et al., 2020). Similarly, in other studies, conducted in Saudi Arabia (Bashir et al., 2017; Bashir et al., 2020) and China (Xia et al., 2019), only male participants were recruited to find the association between nicotine smoking and cognitive function. Another limitation of the present study is that it does not have data of age when smokers started smoking, duration and intensity of cigarette smoking.

Conclusion

We found that young cigarette smokers experienced significant impairment to cognitive function compared to nonsmokers. These results suggest that young adults should quit smoking cigarettes. Cigarette smoking affects cognitive abilities and can trigger demonstrable abnormalities in brain neurocognition.

Supplemental Information

Supplemental Information 1 Raw data

Click here for additional data file.

Additional Information and Declarations

Competing Interests

Author Contributions

Human Ethics

Ethics

Data Availability

The authors declare there are no competing interests.

Tuba Riaz conceived and designed the experiments, performed the experiments, analyzed the data, prepared figures and/or tables, authored or reviewed drafts of the paper, and approved the final draft.

Ghulam Murtaza conceived and designed the experiments, performed the experiments, analyzed the data, prepared figures and/or tables, and approved the final draft.

Areej Arif performed the experiments, authored or reviewed drafts of the paper, and approved the final draft.

Shahid Mahmood, Razia Sultana and Fawaz Al-Hussain analyzed the data, authored or reviewed drafts of the paper, and approved the final draft.

Shahid Bashir conceived and designed the experiments, performed the experiments, analyzed the data, authored or reviewed drafts of the paper, and approved the final draft.

The following information was supplied relating to ethical approvals (i.e., approving body and any reference numbers):

The Institutional Review Board (IRB), University of Gujrat, Pakistan certified that the study fulfilled all ethical standards and did not have any ethical issue (Ref: UOG/ORIC/2019/326).

The following information was supplied relating to ethical approvals (i.e., approving body and any reference numbers):

Institutional Review Board (IRB), University of Gujrat, Pakistan approved this research (UOG/ORIC/2019/326).

The following information was supplied regarding data availability:

Raw data are available in the Supplemental File.

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
