# Peer review of "Nicotine smoking is associated with impaired cognitive performance in Pakistani young people"

_PeerJ, doi:10.7717/peerj.11470_

## Round 0.1 · original submission · Major Revisions

Dr. Murtaza, Thanks for submitting your work to PeerJ. My apologies for the delay but we had some difficulties in recruiting suitable reviewers. I am glad the current reviewers were able to provide detailed feedback on your work and looking forward to your revision.

Hao Chen

Reviewer 1 ·

Basic reporting

The manuscript is reasonably written and reads well. However, the literature review does not satisfactorily align with the topic. There are concerns about some of the articles cited to support the narrative in the manuscript. For example, line 110-112: the cited reference (Bueno, A., Carvalho, F. B., Gutierres, J. M., Lhamas, C., & Andrade, C. M.) refers to the effects of anabolic androgenic steroids (not nicotine) on memory, anxiety, social interaction, acetylcholinestrase activity and oxidative stress in rats. This does not appear to be an appropriate citation for the point raised by the authors. The discussion fails to capture the current understanding on the cognitive enhancing effects of nicotine. Given that this is a controversial topic (cognitive enhancing effects of nicotine),it would be useful to briefly outline both sides of the argument. Furthermore, there is now evidence that the cognitive enhancing effects of nicotine differ on whether nicotine administration is acute or nonacute. (Heishman SJ, Kleykamp BA, Singleton EG. Meta-analysis of the acute effects of nicotine and smoking on human performance. Psychopharmacology (Berl). 2010;210(4):453–469)

Experimental design

The methods are reasonably described. However, in describing the study procedures, important details are left out. For example, were the smokers deprived or allowed to smoke up to and including the time of testing? It is not clear why only males were included in the study. Sampling has not been described and it is unclear how participants were selected. Additionally, a description of the study settings is not adequately provided. The authors report that the study was conducted in the department of zoology at the University of Gujrat without highlighting relevance given than this is a human subjects study.

Validity of the findings

The conclusion appears not to be linked with the results. The study was conducted among male participants and the conclusion should reflect this and not generalize to young adults as presented in the conclusion section.

Additional comments

No comment

·

Basic reporting

References cited in text are, in majority of instances, not matching the sentence they are cited in.
More detailed comments are provided below in General comments.

Experimental design

Authors need to precisely identify and state the knowledge gap and how this research fills that gap.
Methods are not described with sufficient detail (especially how sample was chosen, how investigated scores were calculated etc.).
Authors need to clarify certain ethical considerations.
More detailed comments are provided below in General comments.

Validity of the findings

Almost no demographic data are provided, either in the text of the manuscript or in the provided underlying data. Authors mentioned how they chose the groups so that they do not differ, e.g. in socioeconomic status - but failed to present this anywhere.
Conclusion and implications need to be rephrased in order to be limited to supporting results.
More detailed comments are provided below in General comments.

Additional comments

Line 75: Authors should consider rephrasing the first sentence so that it is more appropriate for a scientific manuscript.
Lines 75-76: The cited references are completely unrelated to the sentence where they are cited - Singh et al. study electronic cigarettes use and Xu et al. study ADHD trends. Please find an appropriate reference and cite it here.
Lines 76-77: Please check whether this information is provided in the cited reference.
Lines 77-78: Again, authors need to check whether this sentence is supported by the cited reference.
Lines 78-81: Please consult the WHO's Report on Global Tobacco Epidemic and cite the appropriate reference.
Line 79: Not just in developed countries, and authors state "more than 5 million deaths per year" globally - shouldn't it be more than 8 million deaths? Please use the most recent data throughout the paper.
General comment: Throughout the entire manuscript, authors need to double check and verify that they have appropriately used and cited the references in the appropriate place in the text, since there are many instances of the opposite being the case.
Line 91: Please add years, you have cited two papers by Benowitz in the list of references - which one is cited here?
Lines 92-94: Please add the appropriate reference.
Overall, it was difficult to check authors' claims since references cited in the text were in most cases completely unrelated to the sentence they are cited in.
Lines 115-116: Authors have cited one study, not studies. Further on, the authors need to specify the population which was included in the cited study.
Introduction: This section needs to give a better and more comprehensive overview on the current body of evidence in this area - description of previous studies and their findings is lacking. This is necessary for the authors to give an adequate explanation of the gap in knowledge their study is trying to address.
Introduction: Describe findings of previous similar studies by sex, to provide a rationale for including only males in the present study.
Line 123: Authors should finish the introduction with a description of the aim or aims of their study.
Lines 127-129: Describe the study population from which the sample is drawn. Specify inclusion and inclusion criteria. How were people recruited for this study?
Lines 129-130: How was a complete clinical history obtained? If a self-reported questionnaire was used - please attach an English version. If medical records of participants were checked to either verify or obtain this information - please state this.
Lines 130-131: Were there no further considerations regarding smokers, in terms of the intensity of smoking?
Lines 131-132: Please clarify what was considered as "never smoked regularly" among the participants in the non-smokers group - does this mean that some of the participants in the "nonsmokers" group could have smoked cigarettes too just not "regularly"?
Lines 133-134: How was the presence of these addictions determined?
Subjects: The study sample description is incomplete. How was the study sample size arrived at? Were groups of 50 smokers and 50 non-smokers large enough to detect a significant difference between them?
Line 138: Did the Institutional Review Board approve the waiver of written consent?
Lines 138-140: Please clarify the ethical considerations - currently, as this text reads, it is not clear whether the ethical approval was received prior to study's beginning?
Line 146: Readers of this article might benefit from a brief description of the CANTAB testing process.
Methods: Add explanations for the scoring of all used tests. For example, authors mention raw and standard score for PRM - this, as well as the scoring for other used tests needs to be explained and supported with appropriate references.
Lines 169-170: Please clarify how were clinical histories collected using the MMSE?
Lines 168-174: Table 1 needs to be cited in this paragraph. But more importantly, Table 1 is supposed to display demographic characteristics of participants - but only provides their sex (when the whole sample is compiled by males?) and age. Table 1 needs to be completely revised and detailed information of participants added to it. This way authors will display whether there really were no differences between the two groups other than their smoking status. Also, smoking habit of the smoking group needs to be presented in detail - was data collected on the smoking start age, duration of smoking, smoking intensity etc.?
Table 2: Indicate used test and p value at which significance was considered in the Table's footnote.
Lines 197-202: None of these studies included young adults - please expand this section by adding comparable studies in terms of the studied population.
Lines 207-209: How was socioeconomic status defined (what did it encompass) and how was it assessed in this study?
Lines 219-222: Results of the present and cited study are not completely similar. Namely, Meo et al. found statistically significantly higher CRT-% in the group of shisha smokers, while the present study found a significantly higher CRT-% in the group of non-smokers. Authors need to discuss these differences. They state in lines 222-223 that he subjects between these two studies differed - please give detail about these differences, since the other study included young men - what were the subjects differences which are mentioned here?
Lines 223-224: Is this sentence really appropriate, since authors have discussed a similar study in the same paragraph, even in previous sentences, a study which also demonstrated he cognitive effect of nicotine smoking using CRT parameters?
Lines 233-234: There are numerous limitations of this study, none of which have been explicitly stated here. Authors should go back to some of the comments given in the section methods, and address those appropriately and if not - add the demonstrated issues to the section limitations.
Lines 237-242: Please limit the conclusion to the findings of the present study and implications for practice based on the results of this study, particularly consider rephrasing the last two sentences so that they specifically correspond to the findings of this study, instead of being a general claim.

---

## Round 0.2 · Minor Revisions

Please make sure you 1) include (as often as appropriate) changes that you made in the rebuttal letter, instead of saying that changes were made, 2) include the responses you provide in rebuttal letter in the manuscript too. Thank you for submitting your work to PeerJ.

Reviewer 1 ·

Basic reporting

Satisfactory

Experimental design

Acceptable

Validity of the findings

Acceptable

·

Basic reporting

Please see comments below.

Experimental design

Please see comments below.

Validity of the findings

Please see comments below.

Additional comments

Thank you for the opportunity to review a revised version of this manuscript. The authors have made many changes and improved the manuscript. However, some questions still need to be addressed.

Lines 231-234: Please provide a reference for the better performance of CANTAB compared to other tests, which is mentioned here.
Aim in the section Introduction: If the intention when planning the study was to include only males, then please revise "Pakistani population" with a more specific description of the studied population.
In addition to the previous comment and in light of revisions made in the Methods section and Discussion, please consider clarifying whether study was planned from the beginning to be conducted only with males, or were there no such limitations but later it proved hard to recruit female participants? Since the questionnaire authors included in the rebuttal letter asked participants about gender, it seems they planned to include females too. Therefore, this needs to be stated as a limitation.
Lines 291-295: If interviewers used questionnaires based on which they collected data from participants face-to-face, please clarify that here when mentioning self-report and interview. Authors mention a questionnaire in their rebuttal letter, and they state they included a questionnaire - but this questionnaire does not cover any of the variables investigated here other than name, gender, age, education, economic status, employment and marital status.
Authors did not respond whether they looked further into the intensity of smoking, instead in their rebuttal letter they repeated the definition of smoking which they used. The question remains - were there no further analyses for smokers who smoked 5 cigarettes a day during the last year and smokers who, e.g., smoked 40 cigarettes a day? It needs to be noted as a limitation of this study if authors did not investigate the age when participants started smoking, duration and intensity of smoking.
In their rebuttal letter, authors state that non-smokers "Nonsmokers (the control group) had never smoked a cigarette in last 30 days." - please use the same definition in the manuscript. Was the same or similar definition used in similar articles? Not in cited paper by Xia et al. for example. What's the rationale behind this definition?
Rebuttal letter includes a response regarding sample size - but this was not incorporated in the manuscript. Were any calculations done based on the cited references of other studies?
Line 297: Please specify that each group consisted of 50 participants, so that it's clear that the entire sample comprised 100 participants.
Finally, please perform a final grammar and spelling check - for example there are instances of words like "Chines" etc.

---

## Round 0.3 · Minor Revisions

Dear Dr. Murtaza,

One of the reviewers suggested a few minor revisions. Please consider revise line 273 to simply state that you were unable to recruit female participants (i.e. remove "due to ..."), which better matches the statement in the methods section. In addition, there are some minor wording changes for your consideration, for example, line 100, remove the second word of the line "software"; and line 139, change word "hard" to "difficult".

Best,

Hao Chen

·

Basic reporting

Please see below.

Experimental design

Please see below.

Validity of the findings

Please see below.

Additional comments

Through the changes they made, authors have made their manuscript more coherent and clear.

English language of the text has been improved, but the authors should recheck the newly added sentences for the last revision, as there are a few instances of missing words (e.g. Lines 260-264), or for example - Line 140 - prevalence is higher not "more".

Important note: Please consider rephrasing the newly added sentence in section Limitations at Line 273 - so that it matches the text given in the Methods section in Lines 139-140 (the Methods text states your inability to include women in this study and the word unable should be underlined in Limitations too).

---

## Round 0.4 · accepted · Accept

Dear Riaz and colleagues:

Thank you for sending back this revision and congratulations on the acceptance of this manuscript for publication in PeerJ.

Hao Chen